# Preacinetobactin not acinetobactin is essential for iron uptake by the BauA transporter of the pathogen *Acinetobacter baumannii*

Lucile Moynié[1,2†], Ilaria Serra[3,4], Mariano A Scorciapino[3,4], Emilia Oueis[5], Malcolm GP Page[6], Matteo Ceccarelli[3], James H Naismith[1,2,7]*

[1]Division of Structural Biology, Wellcome Trust Centre of Human Genomics, Oxford, England; [2]Research Complex at Harwell, Rutherford Laboratory, Didcot, England; [3]Department of Physics, University of Cagliari, Cagliari, Italy; [4]Department of Chemical and Geological Sciences, University of Cagliari, Cagliari, Italy; [5]Biomedical Sciences Research Complex, The University of St Andrews, Scotland, United Kingdom; [6]Department of Life Sciences and Chemistry, Jacobs University, Bremen, Germany; [7]The Rosalind Franklin Institute, Didcot, England

*For correspondence:
naismith@strubi.ox.ac.uk

[†]These authors contributed equally to this work

Competing interests: The authors declare that no competing interests exist.

**Abstract** New strategies are urgently required to develop antibiotics. The siderophore uptake system has attracted considerable attention, but rational design of siderophore antibiotic conjugates requires knowledge of recognition by the cognate outer-membrane transporter. *Acinetobacter baumannii* is a serious pathogen, which utilizes (pre)acinetobactin to scavenge iron from the host. We report the structure of the (pre)acinetobactin transporter BauA bound to the siderophore, identifying the structural determinants of recognition. Detailed biophysical analysis confirms that BauA recognises preacinetobactin. We show that acinetobactin is not recognised by the protein, thus preacinetobactin is essential for iron uptake. The structure shows and NMR confirms that under physiological conditions, a molecule of acinetobactin will bind to two free coordination sites on the iron preacinetobactin complex. The ability to recognise a heterotrimeric iron-preacinetobactin-acinetobactin complex may rationalize contradictory reports in the literature. These results open new avenues for the design of novel antibiotic conjugates (trojan horse) antibiotics.
DOI: https://doi.org/10.7554/eLife.42270.001

## Introduction

*Acinetobacter baumannii* is a Gram-negative bacteria that has been identified by the WHO as a critical priority pathogen for new antibiotic development. *A. baumannii* was highlighted for its increasing role in nosocomial infections and its intrinsic resistance to multiple antibiotics. Several virulence factors have been identified in *Acinetobacter* species including iron-uptake pathways (*Peleg et al., 2012*; *Harding et al., 2018*). Iron is essential for all bacteria, and consequently, human pathogens have to possess an efficient acquisition system to scavenge iron from the host during infection. Gram-negative bacteria secrete small molecules (siderophores), which chelate iron with extremely high affinity. The resulting siderophore-iron complex is recognized by a specific outer-membrane transporter that couples to the TonB system to translocate the complex (and thus iron) into the periplasm where an ABC transporter takes it across the inner membrane (*Noinaj et al., 2010*).

The conserved and essential nature of the uptake system has drawn interest as a potential drug target. Several siderophore-conjugates have been developed to deliver drugs directly to the

periplasm with promising results (*Wencewicz and Miller, 2017*). The successful compounds in part appear able to overcome the resistance of the bacteria to the antibiotic perhaps by increasing the local concentration. This 'trojan horse' strategy is potentially very general and could be used widely to specifically target bacteria and thus avoid the killing of the beneficial microbiome. Three classes of siderophore have been identified as being produced by *A. baumannii,* acinetobactin (*Yamamoto et al., 1994*), fimsbactins (*Proschak et al., 2013*) and baumannoferrins (*Penwell et al., 2015*) and each would be expected to be transported by distinct TonB-dependent transporters. Analogues of the bis-catechol siderophore fimsbactin A linked to antibiotics such as loracarbef/ciprofloxacin (*Wencewicz and Miller, 2013*), daptomycin (*Ghosh et al., 2018*) and cephalosporin (*Liu et al., 2018*) have proven effective in killing *A. baumanni. A. baumanni* has been also targeted by directly inhibiting the biosynthesis of acinetobactin (*Drake et al., 2010*; *Neres et al., 2013*) and by disrupting the acinetobactin uptake mechanism through the use of an oxidized version of acinetobactin (*Bohac et al., 2017*).

An important requirement for rational development of a trojan horse strategy is understanding the chemical nature of the siderophore and the structure activity relationship of its translocation, that is what parts are recognised and where cargo can be attached. The acinetobactin system is of particular interest since it has been identified as a virulence factor in mouse infection models (*Gaddy et al., 2012*). Acinetobactin is synthesized as a precursor, preacinetobactin, which rearranges non-enzymatically and irreversibly at pH >7 to give acinetobactin (*Figure 1A*) (*Sattely and Walsh, 2008*; *Wuest et al., 2009*; *Shapiro and Wencewicz, 2016*). There have been different reports on whether acinetobactin or preacinetobactin or both are transported into the bacteria (*Song et al., 2017*; *Shapiro and Wencewicz, 2017*) complicating rational design. The *A. baumannii* outer-membrane transporter protein, BauA, that recognises this siderophore has itself been identified as a vaccine candidate (*Esmaeilkhani et al., 2016*).

We report the structure of the BauA transporter bound to its cognate siderophore. Our structure reveals the molecular basis of recognition and sheds light on the question of which isomer is recognised. We establish that BauA recognises preacinetobactin not acinetobactin. Interestingly, we actually found that a mixed complex of both preacinetobactin and acinetobactin can be bound to the transporter. Under the most likely physiological conditions, low iron and neutral pH, the mixed species would seem the most plausible compound for uptake. Our data provide structural insights that can be used to guide rational design of 'trojan horse' antibiotics against *A. baumannii.*

## Results

### Structural biology

The apo structure of BauA was solved to a resolution of 1.8 Å by single wavelength anomalous diffraction (SAD) using selenomethionine-labeled proteins. Residues 33 to 703 of the protein are well defined in the electron density map, while the *N*-terminal region comprising the TonB box is missing, presumably because it is flexible. BauA has the typical outer-membrane TonB-dependent transporter (TBDT) fold with a β-barrel formed by 22 antiparallel strands, several extracellular loops, and a plug domain folded inside of the barrel (*Figure 1B*). Compared to the other TBDTs, the extracellular loops of the β-barrel form a large open cavity bordered by mainly short and apolar amino acids giving large access to the loops NL1-NL3 of the plug domain (*Figure 1—figure supplement 1*). Search of the structural database reveals the closest structural homologues are ferrichrome (*Locher et al., 1998*) (1by5, rmsd of 2.5 Å for 634 residues aligned and 2fcp, 2.4 Å for 631 residues) and enantiopyochelin (*Brillet et al., 2011*) (3qlb, rmsd of 2.5 Å for 630 residues) transporters. The asymmetric unit contains three monomers arranged around a threefold symmetry axis reminiscent of the porin arrangement (*Figure 1—figure supplement 2A*) and thus plausibly biologically meaningful. PISA (*Krissinel and Henrick, 2007*), a computational assessment of oligomeric state in the crystal, suggests a trimer with a high degree of confidence. Biophysical methods (gel filtration and light scattering) do not support a trimeric assembly. Size Exclusion Chromatography coupled to Multi-Angle Light Scattering (SEC-MALS) analysis in C8E4 is consistent with monomer whilst native gel shows the protein runs more closely to a dimer; however, the detergent bound to the protein may increase the apparent weight suggesting a monomer in solution (*Figure 1—figure supplement 2B*; *Figure 1—figure supplement 2C*).

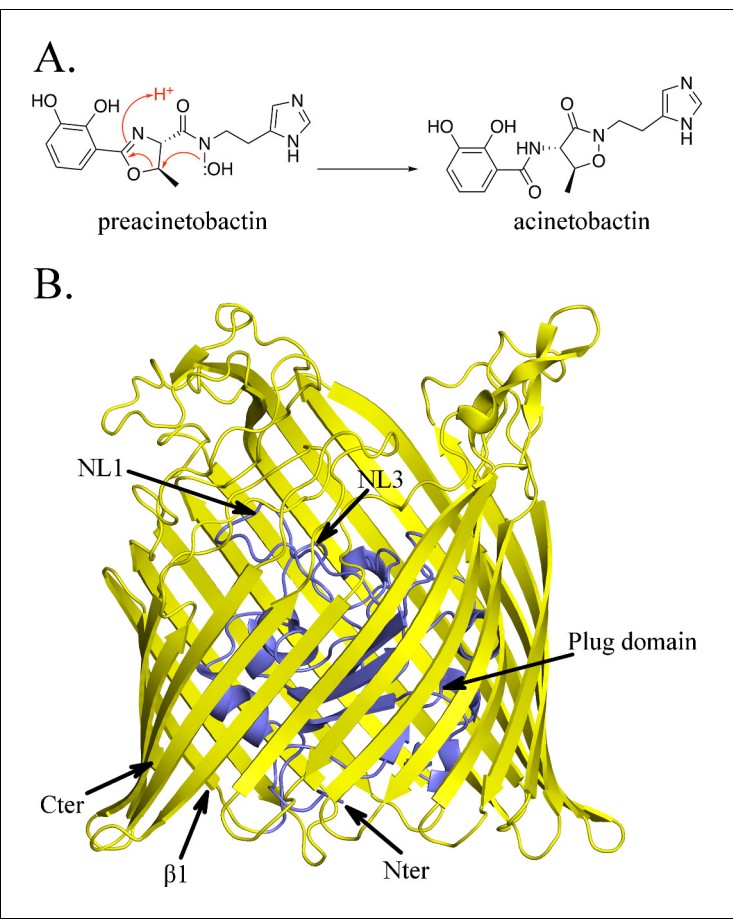

**Figure 1.** Structure of BauA, the (pre)acinetobactin transporter in *A. baumannii*. (**A**) Schematic representation of the spontaneous isomerization of preacinetobactin into acinetobactin occurring at pH >7. (**B**) Overall structure of BauA. The N-ter plug domain is colored in blue and the β-barrel in yellow.

DOI: https://doi.org/10.7554/eLife.42270.002

The following figure supplements are available for figure 1:

**Figure supplement 1.** Surface comparison of BauA (**A**), FepA (**B**) and FpvA (**C**).
DOI: https://doi.org/10.7554/eLife.42270.003
**Figure supplement 2.** BauA oligomer state.
DOI: https://doi.org/10.7554/eLife.42270.004
**Figure supplement 3.** Structure-based sequence alignment of BauA with other siderophore transporters.
DOI: https://doi.org/10.7554/eLife.42270.005

We incubated separate native protein crystals with preacinetobactin and acinetobactin compounds (both compounds preloaded with $Fe^{3+}$); surprisingly in both cases, we obtained the same structure: BauA bound to a heterocomplex $Fe^{3+}$-preacinetobactin-acinetobactin (1:1:1) (*Figure 2A*). Given that preacinetobactin is the precursor for acinetobactin in a pH-dependent irreversible rearrangement reaction, it would explain this observation only for the preacinetobactin sample. LC-MS analysis of preacinetobactin in the crystallization conditions shows that in the absence of iron, the majority of preacinetobactin is converted to acinetobactin in 30 min and no preacinetobactin was observed the next day (*Figure 3—figure supplement 1C*); indicating that the presence of iron stabilizes preacinobactin. On the other hand, we discovered that there was a small amount (~2%) of preacinetobactin in our acinetobactin sample (*Figure 3—figure supplement 1A*). After full conversion of preacinetobactin into acinetobactin (*Figure 3—figure supplement 1B*), no crystal structure complex with BauA was obtained with this material.

The coordination of the iron is octahedral with four coordination atoms from preacinetobactin (one hydroxy group of the catecholate, the hydroxy of the hydroxamate, the nitrogen of the methyl

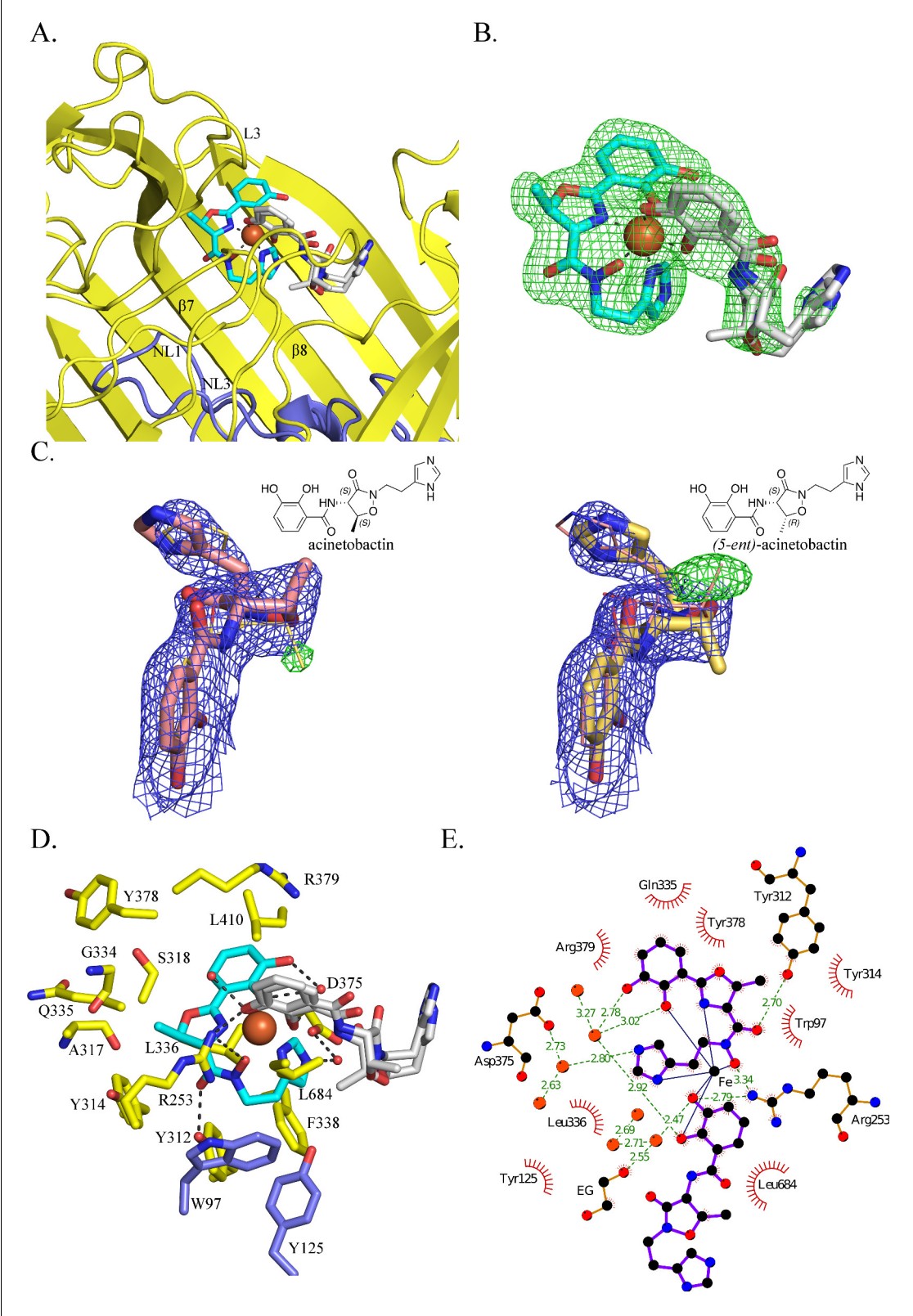

**Figure 2.** BauA binds a heterocomplex $Fe^{3+}$, preacinetobactin and acinetobactin. (**A**) Overall structure of BauA in complex with $Fe^{3+}$, preacinetobactin and acinetobactin. The N-ter plug domain is colored in blue and the β-barrel in yellow. Preacinetobactin is shown as sticks with carbon atoms colored cyan and acinetobactin carbon atoms are in white, nitrogen in blue and oxygen in red. The $Fe^{3+}$ is represented as an orange sphere. Secondary structure elements involved in the binding site have been labeled. (**B**) $F_O$-$F_C$ electron density from the original omit map at 3 σ around $Fe^{3+}$,

*Figure 2 continued on next page*

*Figure 2 continued*

preacinetobactin and acinetobactin complex. The color scheme of the $Fe^{3+}$- siderophores is the same as A. We model the second molecule as a mixture of two diastereomers of acinetobactin. (C) $2F_O$-$F_C$ (blue) and $F_O$-$F_C$ (green) electron density maps at 1 and 3 σ, respectively, around acinetobactin. In the left panel, only the acinetobactin (carbon atom represented in pink) has been added in the model before refinement. Position of (*5-ent*)-acinetobactin is shown in yellow line for information. In the right panel, only (*5-ent*)-acinetobactin (yellow stick) has been added in the model. Position of acinetobactin is shown in pink line for information. Schematic representation of both compounds has been added in the top corner for each image. (D) Binding site of the siderophores. Residues within 4.0 Å of the siderophores are displayed and hydrogen bonds are shown as black broken lines. Carbon atoms of residues of the β-barrel are in yellow and the ones of the plug domain in blue. (E) Ligplot diagram of $Fe^{3+}$, preacinetobactin, acinetobactin bound to BauA. Covalent bonds of the siderophore and protein residues are in purple and brown sticks, respectively. Hydrogen bonds are represented by green dashed line and hydrophobic contacts are shown as red semi-circles with radiating spokes. One molecule of ethylene glycol is present in the binding site. Figure prepared with Ligplot (*Wallace et al., 1995*).

DOI: https://doi.org/10.7554/eLife.42270.006

The following figure supplements are available for figure 2:

**Figure supplement 1.** The location of BauA binding site is similar to the ones of the other siderophore transporters.

DOI: https://doi.org/10.7554/eLife.42270.007

**Figure supplement 2.** The second molecule has been modeled as a mixture of two diastereomers of acinetobactin.

DOI: https://doi.org/10.7554/eLife.42270.008

**Figure supplement 3.** Model of acinetobactin, preacinetobactin, (*5-ent*)-preacinetobactin binding.

DOI: https://doi.org/10.7554/eLife.42270.009

oxazoline and one nitrogen (N1) of the imidazole group) and two hydroxyls from acinetobactin (the catecholate) (*Figure 2B*). The $Fe^{3+}$-preacinetobactin coordination geometry is similar to that observed for one $Ga^{3+}$ in the gallium complex with anguibactin but the later forms a $Ga_2L_2MeOH_2$ complex with two methanol molecules bridging the two four coordinate $Ga^{3+}$ ions (*Hossain et al., 1998*). The net charge of the $Fe^{3+}$-preacinetobactin complex assuming three deprotonated catecholates, one deprotonated hydroxamate and $Fe^{3+}$ would be −1, consistent with stabilization by Arg253. The interaction between the catecholate and the nitrogen of the methyl oxazoline of the acinetobactin and the N1 of the imidazole group and the catecholate of preacinetobactin may play a role in the stabilization of the complex. A water molecule bridges the catecholate oxygen atoms of preacinetobactin.

The protein interacts mainly with the preacinetobactin molecule. The binding site is located inside the β-barrel similar to other siderophore complexes (*Locher et al., 1998*; *Brillet et al., 2011*; *Chimento et al., 2005*; *Cobessi et al., 2005a*; *Cobessi et al., 2005b*; *Ferguson et al., 1998*; *Ferguson et al., 2002*) (*Figure 1—figure supplement 3*; *Figure 2—figure supplement 1*; *Figure 2A*). The siderophores interact mainly with the residues of the β-barrel and the extracellular loops. BauA recognises the preacinetobactin through two direct hydrogen bonds between the hydroxamate group and Tyr312 of β7 and Arg253 of L3 and one interaction through a bridging water molecule between Asp375 and the nitrogen N2 of the imidazole, respectively (*Figure 2D and E*). The binding site is predominantly hydrophobic in character with van der Waals interactions involving residues Trp97, Tyr125, Tyr314, Tyr378, Gly334, Gln335, Leu336, Arg379 and Leu684. Although there are no hydrogen bonds between the siderophores and the plug domain, two residues of the plug domain (Trp97 and Tyr125) do make van der Waals interactions with preacinetobactin. The acinetobactin molecule makes only one hydrogen bond (Arg253) and one van der Waals interaction (Leu684). There are no changes in the TonB region as a result of ligand binding. The experimental electron density is most convincingly fitted by a mixture of two diastereomers of acinetobactin (*Figure 2C*), in an earlier co-crystalliszation experiment the 'wrong' diastereomer of acinetobactin was clearer (*Figure 2—figure supplement 2F*; *Figure 2—figure supplement 2G*). Consequently, we disfavor explanations for the electron density based on disorder or a mixture of acinetobactin/preacinetobactin (*Figure 2—figure supplement 2*). However, we find no evidence for the 'wrong' diastereomer in any NMR spectra of the starting materials which all match the literature report for the 'correct' diastereomer (*Shapiro and Wencewicz, 2016*; *Takeuchi et al., 2010*). Spontaneous chemical interconversion of the diastereomers seems implausible in simple solution. Thus, we are left to suggest the presence of small contaminating amount of the 'wrong' diastereomer or chemical conversion under crystallization conditions (protein, buffer, salts, crystal effects etc).

## Siderophore binding

Three observations led us to speculate BauA preferentially recognises preacinetobactin; first, the protein ligand contacts are almost all with the preacinetobactin molecule, secondly the reduction of the preacinetobactin impurity to undetectable level gave no co-complex, and finally even a small amount of preacinetobactin as an impurity lead to complex formation. To test our hypothesis, we conducted isothermal titration calorimetry (ITC) experiments with both compounds (*Figure 3A* and *Figure 3—figure supplement 2*). These results are unambiguous: the $Fe^{3+}$-preacinetobactin binds to BauA with nM affinity whilst $Fe^{3+}$-acinetobactin shows no binding. The titration curves and thus affinity for the $Fe^{3+}$-preacinetobactin complex and BauA differ depending on the ratio of $Fe^{3+}$ to siderophore. A stoichiometry of 1:1 $Fe^{3+}$: preacinetobactin has a Kd of 763 nM, whereas 1:2 $Fe^{3+}$: preacinetobactin bind with 83 nM affinity. In both cases, the stoichiometry of binding was one $Fe^{3+}$

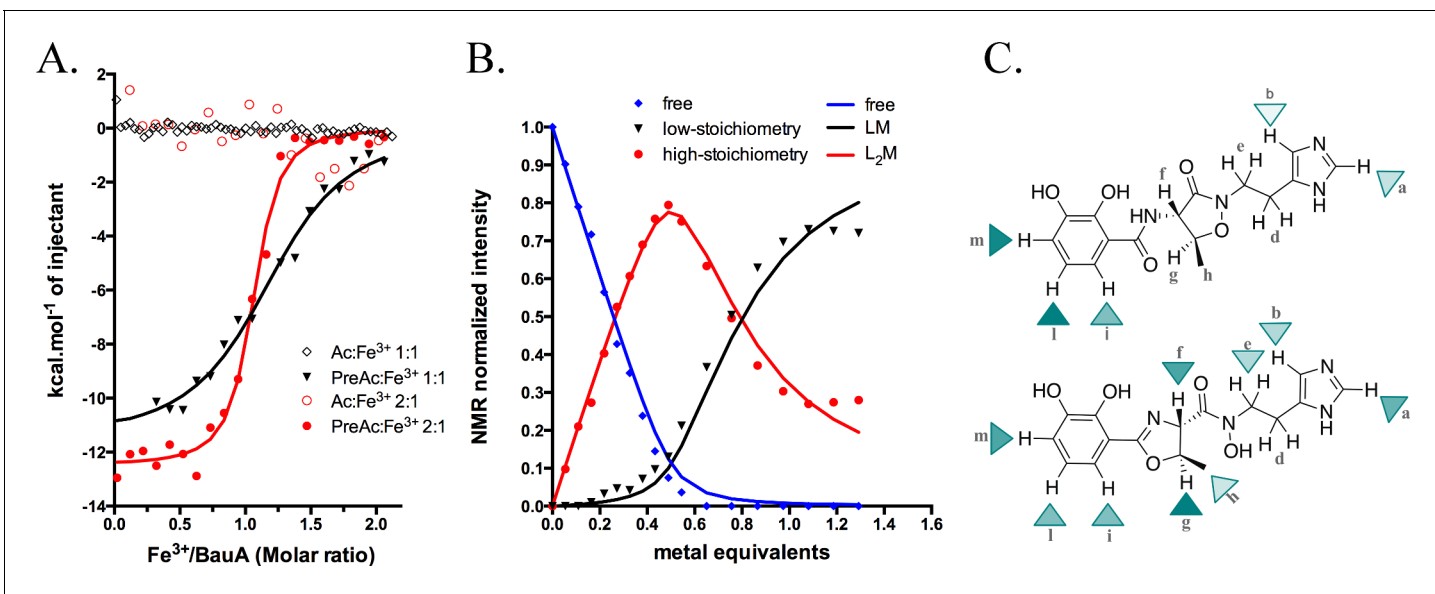

**Figure 3.** Siderophore iron complex stoichiometry. (A) Isothermal calorimetry titrations of $Fe^{3+}$-preacinetobactin (ratio 1:2 in red sphere and 1:1 in black triangle) and $Fe^{3+}$-acinetobactin (ratio 1:2 in open red circle and 1:1 in open black lozenge). For $Fe^{3+}$-preacinetobactin, the isotherm fitted with the Origin software gave a Kd of 83 and 763 nM for the ratio 1:2 and 1:1, respectively. With both ratios, no interaction has been detected for $Fe^{3+}$-acinetobactin. (B) Results of curve fitting for titration of preacinetobactin in acetate buffered saline (ABS) at pH 5.0 (meter reading) with $Ga^{3+}$. Results are shown for the proton resonance due to the methyl group and the curves correspond to the complexation model with the lowest total square deviation between the calculated (solid lines) and the experimental data (dots). Key: L, ligand; M, metal. (C) Molecular structure of acinetobactin (top) and preacinetobactin (bottom) is shown together with color-coded markers. Cyan triangles were used to represent the chemical shift difference observed in ABS between the resonance of the $L_2M$ complex and the corresponding one for the free L in the absence of metal ions (*Table 1*). Maximum colour intensity was set for the largest shift observed for each case and rescaling the other markers proportionally. Some markers are missing because corresponding resonances were not sufficiently resolved.

DOI: https://doi.org/10.7554/eLife.42270.010

The following figure supplements are available for figure 3:

**Figure supplement 1.** Preacinetobactin and acinetobactin analyses by HPLC.

DOI: https://doi.org/10.7554/eLife.42270.011

**Figure supplement 2.** Isothermal calorimetry titrations of $Fe^{3+}$-Preacinetobactin (ratio 1:1 (A) and 1:2 (B)) and $Fe^{3+}$-acinetobactin (ratio 1:1 (C) and 1:2 (D)) with BauA.

DOI: https://doi.org/10.7554/eLife.42270.012

**Figure supplement 3.** Selected [1]H-NMR spectra along the titration of preacinetobactin and acinetobactin with $Ga^{3+}$.

DOI: https://doi.org/10.7554/eLife.42270.013

**Figure supplement 4.** Preacinetobactin is not converted to acinetobactin when involved in metal ion coordination.

DOI: https://doi.org/10.7554/eLife.42270.014

**Figure supplement 5.** [1]H NMR spectral data of preacinetobactin (A) and acinetobactin (B).

DOI: https://doi.org/10.7554/eLife.42270.015

to one protein; in neither case did we see any evidence for a biphasic recognition model that has been proposed between enterobactin and FepA (*Payne et al., 1997*).

## Siderophore-metal complex stoichiometry

The different affinities of $Fe^{3+}$-preacinetobactin observed in ITC prompted us to examine the stoichiometry of the complex. Different molar ratios of metal to preacinetobactin were examined through $^{1}$H-NMR by using $Ga^{3+}$ as a diamagnetic alternative to $Fe^{3+}$ (*Matzanke, 2006*) (*Figure 3—figure supplement 3*). The conversion from preacinetobactin to acinetobactin is much slower at the lower pH used (pH 5) (*Shapiro and Wencewicz, 2016*), therefore on the timescale of the experiment, we ignored the small amount of acinetobactin. The titration reveals complexation of the free siderophore in the presence of low amounts of metal to give a complex of two siderophores to one metal, with the free siderophore almost disappearing when the metal reaches 0.5 molar equivalent (*Figure 3B*). Two different new sets of resonances were observed, both with chemical shift distinct from the free siderophore. This magnetic inequivalence clearly indicates two different ion coordination modes for each molecule of preacinetobactin in the $ML_2$ complex. The only exception was the ethyl-imidazole group, which showed one set of resonances the same as free siderophore as well as a new series of different resonances. This is consistent with the imidazole being directly involved in metal coordination in one of the siderophores whilst in the other siderophore the imidazole is uncoordinated. As the amount of metal is further increased, a 1:1 complex of the siderophore with the metal appeared whilst the amount of the 2:1 complex decreased. This indicates there is an equilibrium in which the $ML_2$ complex is favored at low metal concentration but the ML complex is predominant at higher metal concentrations. The formation constants were calculated as $K_f(ML)=1.2 \cdot 10^5$ $M^{-1}$; $K_f(ML_2)=7.2 \cdot 10^3$ $M^{-1}$. Beyond 1.0 eq. $Ga^{3+}$, a noticeable deviation between the model and the experimental data can be seen (*Figure 3B*). This is plausibly due to a fraction of acinetobactin present, which can coordinate some of the ML complexes formed by preacinetobactin giving the mixed complex seen in the structure. Preacinetobactin in these mixed complexes would be magnetically undistinguishable from those in the true $ML_2$ complexes, thus artificially inflating the apparent concentration of this species.

The experiment was repeated with acinetobactin, however, titration was not possible beyond 0.35 molar equivalent of metal due to precipitation. We were able to observe the formation of a complex with concomitant decrease in free siderophore. The decrease in free siderophore versus ion concentration (up to 0.35) is more pronounced for preacinetobactin. NMR shows that metal binding to acinetobactin results in the largest shifts of signals within the catechol group, whereas in preacinetobactin the largest shifts are observed in the central portion of the molecule, where the hydroxamate moiety is located in agreement with the crystal structure (*Figure 3C* and *Table 1*).

## Discussion

The nature of one of the main siderophore molecules taken up by *A. baumannii* is important, both for understanding the physiology of this significant pathogen and for the rational design of the so-called 'trojan horse' antibiotics. Conflicting reports are extant in the literature as to whether the molecule which is taken up is preacinetobactin or acinetobactin (*Song et al., 2017*; *Shapiro and Wencewicz, 2017*). As our own experiments have shown, small amounts of impurities that are hard to eliminate coupled to ready conversion of preacinetobactin to acinetobactin poses significant technical challenges in resolving this problem.

The siderophore is known to be bound from the medium by the integral outer-membrane BauA protein. Structural biology strongly implies that BauA in fact recognizes preacinetobactin and not acinetobactin. In the crystal structure, preacinetobactin fits into a complementary pocket; a simple molecular model shows that the same site would not bind acinetobactin due to extensive van der Waals clashes with Tyr312 (*Figure 2—figure supplement 3B*). Preacinetobactin wraps around the metal ion by occupying four out of the six possible coordination sites. ITC confirms this hypothesis and demonstrates very tight binding for the $Fe^{3+}$-preacinetobactin complex but none at all for acinetobactin; thus, for transport, preacinetobactin must be present. We note that our crystallization results show that even small amounts of contaminating preacinetobactin present in our original acinetobactin sample (detected only by careful chemical characterization after the fact), will bind to the protein, a reflection of the large difference in affinity between preacinetobactin and acinetobactin.

**Table 1.** Chemical shift difference (absolute value of ppm in the absence – ppm in the presence of metal ions) is reported for the resonances of both preacinetobactin and acinetobactin during titration.

| $^1$H nuclei[†] | preacinetobactin[‡] | Acinetobactin |
|---|---|---|
| a | 0.34 | 0.07 |
| b | 0.17 | 0.02 |
| d | n.r. | n.r. |
| e | 0.16 | n.r. |
| f | ~0.4[§] | n.r. |
| g | 0.56 | n.r. |
| h | 0.12 | n.o. |
| i | 0.27 | 0.15 |
| l | 0.27 | 0.30 |
| m | 0.39 | 0.28 |

The titration performed in acetate buffered saline with $Ga^{3+}$. Reported values correspond to 0.5 and 0.3 eq. $Ga^{3+}$ in the case of preacinetobactin and acinetobactin, respectively.[†] Labels refer to **Figure 3C**.[‡] Two resolved resonances were observed for protons g, h, i, l and m, due to two different $L_2M$ species (see the main paper). Average values are reported here.[§] Complex overlapped with residual water's resonance. Value estimated by comparison with additional titration performed in DMSO. n.r.: not sufficiently resolved. n.o.: difference not observed.

DOI: https://doi.org/10.7554/eLife.42270.016

We observed that in the absence of $Fe^{3+}$, preacinetobactin rapidly converts to acinetobactin under the crystallization conditions. In our hands, the $Fe^{3+}$ complex stabilized preacinetobactin by preventing the conversion to acinetobactin. This was also observed by NMR in solution with $Ga^{3+}$ (**Figure 3—figure supplement 4**) and it agrees with the literature (**Shapiro and Wencewicz, 2016**).

We observed a heterotrimeric $Fe^{3+}$-preacinetobactin-acinetobactin complex in the crystal. The acinetobactin molecule is not itself recognised by the protein, rather it is the favorable coordination of the catechol group to the remaining two sites of $Fe^{3+}$ that drives complex formation. The lack of recognition would seem supported by our observation that in the crystal structure there is an unnatural diastereomer of acinetobactin playing the role of the second ligand (**Figure 2—figure supplement 2**; **Figure 2C**). In contrast, there was no electron density for the other diastereomer of preacinetobactin at the primary site consistent with simple modelling showing it would clash with the protein (**Figure 2—figure supplement 3C**). The second molecule in the complex, acinetobactin, has arisen from chemical conversion of preacinetobactin, a conversion known to occur rapidly in solution. We have no reason to predict that the conversion is in anyway affected by the crystal environment. The presence of preacinetobactin in the crystal is consistent with its chemical stabilization by $Fe^{3+}$. Modeling suggests that the catechol of a second molecule of preacinetobactin would be able to bind to the unoccupied iron sites via its catechol without any clash with the protein (**Figure 2—figure supplement 3D**). We attribute the fact we do not observe it in preacinetobactin-soaked crystals as a consequence of the rapid conversion of catechol only ligated preacinetobactin into acinetobactin.

Based on the $^1$H-NMR experiment performed with $Ga^{3+}$, we predict that at siderophore/iron ratio >0.8, the dominant complex is $Fe^{3+}$-preacinetobactin ($Fe^{3+}L$) whilst at siderophore/iron ratio <0.6, the $Fe^{3+}L_2$ is dominant. ITC suggests that the $Fe^{3+}L_2$ binds to BauA with higher affinity than the $Fe^{3+}L$ complex (**Figure 3—figure supplement 2**; **Figure 3A**).

Combining all our data, we suggest that under physiological conditions such as blood where free iron is very low, it is the heterotrimeric $Fe^{3+}$-preacinetobactin-acinetobactin complex that is most likely to be bound by BauA and taken up (**Figure 4a**). Only under acidic conditions (pH <6), where preacinetobactin conversion is slower, is it conceivable that the $Fe^{3+}$- (preacinetobactin)$_2$ complex could be taken up. This maybe significant as *A. baumannii* infections do occur in an acidic environment (**Peleg et al., 2008**; **Higgins et al., 2010**). Our data cannot rule out the possibility that during uptake, the second ligand, bound only by the catechol group, is removed immediately before

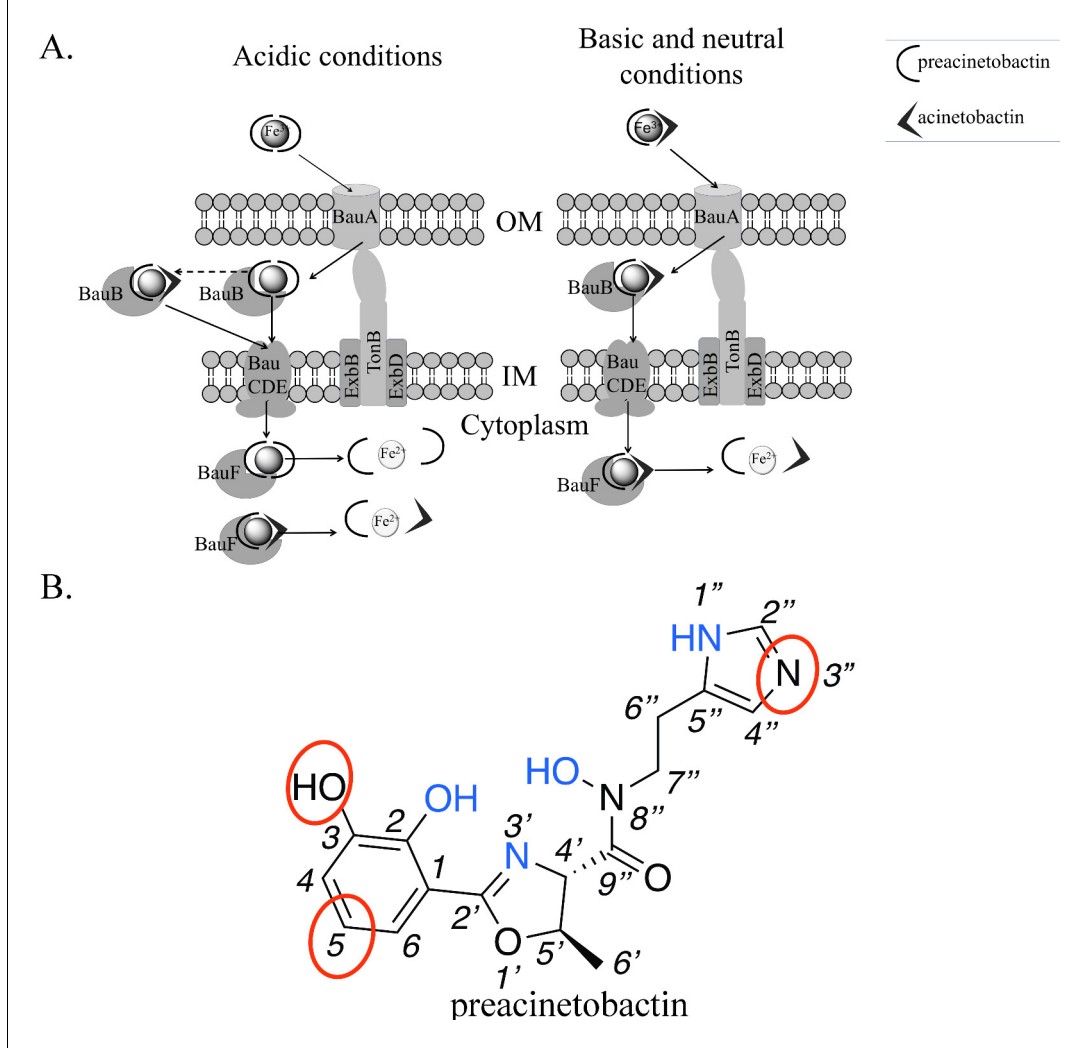

**Figure 4.** Model of preacinetobactin/acinetobactin uptake and opportunities for design. (**A**) In acidic environments, preacinetobactin is stable. Two molecules can chelate $Fe^{3+}$ and the complex could be stable thus recognised and transported by the TonB dependent transporter, BauA. Once in the periplasm, BauB, a periplasmic binding protein, delivers the $Fe^{3+}$-siderophore to the inner membrane ABC transport system (BauCDE) to cross the inner membrane. The release of the Fe occurs into the cytoplasm with the help of BauF, a hydrolase. In neutral pH conditions, preacinetobactin isomerizes into acinetobactin. $Fe^{3+}$ chelation stabilizes one molecule of preacinetobactin but the second molecule isomerizes. The heterotrimeric $Fe^{3+}$-preacinetobactin-acinetobactin complex is recognised and possibly transported by BauA. (**B**) Functional groups involved in the $Fe^{3+}$ chelation are highlighted in blue and the ones which could be modified in the design of siderophore antibiotic conjugates are circled in red.

DOI: https://doi.org/10.7554/eLife.42270.018

translocation. In this scenario only preacinetobactin would be transported but could convert to acinetobactin after transport when iron has been removed. However, in the light of the tendency to form stable $ML_2$ complexes, removal of one L molecule is expected to be energetically unfavorable. The definitive resolution of this question is beyond the scope of this study.

Our data would appear to resolve the controversy in the literature as to the nature of siderophore taken up by BauA transporter. A complex containing both preacinetobactin and acinetobactin will bind to BauA and may in fact be taken up, but the presence of preacinetobactin is essential. Our work does not eliminate the possibility of another TBDT transporting acinetobactin, but so far none has been identified in *A baumannii*. It will be interesting to study whether the BauA transporter when it recognises preacinetobactin can 'pull' through catechols other than acinetobactin which bind to the two vacant iron coordination sites. The stabilization of preacinetobactin by formation of

the tetradentate $Fe^{3+}$ complex resolves the puzzle as to how a chemically unstable molecule is transported.

The structure outlines two new routes for the rational design of siderophore conjugates that could be taken up via BauA. The first approach is modification of preacinetobactin. The crystal structure suggests the uncoordinated nitrogen atom of the imidazole ring has the potential for modification without affecting recognition. Similarly, one of the oxygens and a carbon of catechol seem plausible candidates for expansion (*Figure 4b*). *Shapiro and Wencewicz (2017)* and *Song et al. (2017)* reported important indications about the role of the different portions of both ligands by comparing the biological activity of multiple derivatives and checking complex formation through UV-vis spectroscopy. These studies confirm the 2-OH of preacinetobactin is essential but changes at the 3 and 5 positions of the catechol are well tolerated (no information on the four position was reported). The second approach would be to design a catechol antibiotic conjugate that forms a high-affinity heterotrimeric complex with iron and preacinetobactin. The structure suggests that there is no protein imposed recognition requirement on the second molecule, this potentially unlocks a diversity of conjugate molecules complementing the fimsbactin A type conjugates (*Wencewicz and Miller, 2013*; *Ghosh et al., 2018*; *Liu et al., 2018*). The third approach would be the design of a molecule that hexacoordinates the iron molecule, whilst preserving the protein preacinetobactin contacts thus binding to BauA. This compound with its much higher affinity would easily outcompete preacinetobactin for iron and could be designed so as to block translocation by BauA.

# Materials and methods

## Preparation of the substances

Acinetobactin and preacinetobactin were synthesized following the synthetic schemes and the protocols described by *Takeuchi et al., 2010*. The collected $^1$H-NMR data were similar to the $^1$H-NMR described by *Takeuchi et al. (2010)* and Shapiro *et al.* (*Shapiro and Wencewicz, 2016*; *Shapiro and Wencewicz, 2017*)(*Figure 3—figure supplement 3 and 5*).

## BauA cloning

Position of the cleavage of the signal peptide of the proteins was predicted with Signal P4.0 (*Petersen et al., 2011*). The coding sequence of the mature protein BauA (uniprot Q76HJ9) was amplified from the genomic DNA ATCC 19606 using the primers 5'- AGGGGCGCCATGGCTGTTA TTGATAATTCAACAAAAACTC and 5'- AATTTGGATCCTTAAAAGTCATATGATACAGATAGCATA TACG and Pfu DNA polymerase. PCR product has been digested by NcoI and BamHI restriction enzymes. The gene was cloned with an *N*-terminal tobacco etch virus (TEV) protease cleavable His$_7$-tag in pTAMAHISTEV vector using NcoI and BamHI restriction sites.

## Over-expression in *E. coli* and purification

The proteins were over-expressed in *E. coli* C43 (DE3) cells. Cells were grown at 37°C in LB medium containing 100 μg.ml$^{-1}$ ampicillin until an OD$_{600}$ of 0.7 and then induced with 0.4 mM IPTG at 25°C overnight. The proteins were purified following the protocol used for *Ab*PirA as described in *Moynié et al. (2017)*. Isolated outer-membrane pellets were solubilized with 7% octylpolyoxyethylene (octylPOE)) and the proteins were purified with a Ni$^{2+}$–NTA affinity chromatography followed by the His-tag cleavage by TEV protease and a second IMAC column. Finally, the protein was loaded on a Superdex S200 gel filtration chromatography (10 mM Tris pH 7.5, 150 mM NaCl, 0.45% C8E4) and concentrated to 10 mg.ml$^{-1}$ before setting up the crystallization plates. Selenomethionine-substituted proteins were produced with the SelenoMethionine Medium Nutrient Mix (Molecular Dimension) in C43(DE3) cells according to the feedback inhibition method (*van den Ent et al., 1999*). Purification of the protein was the same as the native but with 5 mM β-mercaptoethanol added in all purification buffer.

## Structural biology

Crystals of BauA appeared at 20°C after a few days by mixing 1 μl of protein solution (10 mg.ml$^{-1}$) with 1 μl of reservoir solution containing 10% PEG 8000, 0.1 M Hepes pH 7.5, 0.1M KCl, and 10% ethylene glycol. Crystals were frozen with the same solution containing 26% ethylene glycol.

Complex structures were obtained by soaking apo crystals for few hours with mother liquor containing 5 mM $Fe^{3+}$-acetylacetonate siderophore (preacinetobactin or acinetobactin) in 1:1 ratio ($Fe^{3+}$: siderophore) before cryoprotection. Subsequent LC-MS analysis showed that the acinetobactin sample had a small amount (~2%) of preacinetobaction present. Preacinetobactin was fully converted overnight into acinetobactin in a 150 mM sodium phosphate pH 8 solution before adding $Fe^{3+}$ (1:1 ratio) with a ferric ammonium citrate solution (the colour changes from red to blue in 30 min). The ferric complex solution was diluted with the mother liquor to a final concentration of 2 mM before soaking the crystal overnight. No complex has yet been obtained.

Data were collected at the beamlines i02, i03 and i24 at the Diamond light source Oxfordshire. Data were processed with XIA2 (*Winter, 2010*; *Zhang et al., 2006*; *Sauter et al., 2004*; *Kabsch, 1993*; *Evans, 2006*). Structure of BauA was solved by SAD with the program AUTO-RICKSHAW (*Panjikar et al., 2009*). Structures of the complexes have been solved using the apo structure. Models were adjusted with COOT (*Emsley and Cowtan, 2004*) and refinement was carried out using REFMAC in the CCP4 program suite with TLS parameters (*Murshudov et al., 1997*). Coordinates and topologies of ligands were generated by PRODRG (*Schüttelkopf and van Aalten, 2004*). Final refinement statistics are given in *Table 2*. Atomic coordinates and structure factors have been deposited in the Protein Data Bank (6H7F, 6H7V, 6HCP). The quality of all structures was checked with MOLPROBITY (*Chen et al., 2010*). Figures were drawn using PYMOL (*Schrodinger, 2010*).

## Isothermal microcalorimetry titration

Affinity of BauA for $Fe^{3+}$-preacinetobactin and $Fe^{3+}$-acinetobactin were measured by isothermal titration calorimetry using a VP-ITC or a ITC200 instrument (GE Healthcare) at 25 °C. Ratio 1:2 and 1:1 of $Fe^{3+}$: preacinetobactin or acinetobactin have been tested. Solutions of 100 mM of compound and either 100 mM or 50 mM of $Fe^{3+}$-acetylacetonate in DMSO were diluted with the dialysis buffer (50 mM Sodium phosphate pH 7.5, 50 mM NaCl, 0.8% OctylPOE). Titrations of preacinetobaction were performed using 2 μl injections of 100 μM of $Fe^{3+}$ and either 100 μM or 200 μM preacinetobactin into 10 μM BauA.

Several conditions have been tested for $Fe^{3+}$-acinetobactin. Titrations of acinetobaction were performed using 2 μl injections of 100 μM of $Fe^{3+}$ and 200 μM acinetobactin (1:2) into 10 μM BauA. Titrations of acinetobaction were also performed with higher concentration using 5 μl injections of 370 μM of $Fe^{3+}$-acinetobactin (1:1) in 20 μM BauA in the same buffer. The heats of dilution were measured by injecting the ligands into the buffer. Titration curves were fitted using Origin software.

## LCMS analysis

The HPLC grade acetonitrile (MeCN) was purchased from Fisher. Aqueous buffers and aqueous mobile-phases for HPLC were prepared using water purified with an Elga Purelab Milli-Q water purification system (purified to 18.2 MΩ.cm) and filtered over 0.45 μm filters.

Analytical RP-HPLC-MS was performed on an Agilent infinity 1260 series equipped with a MWD detector using a Macherey-Nagel Nucleodur C18 column (10 μm x 4.6 × 250 mm) and connected to an Agilent 6130 single quad apparatus equipped with an electrospray ionization source using the following chromatographic system: 1 mL/min flow rate with MeCN and 0.1% TFA in $H_2O$ [95% TFA/$H_2O$ (5 min), linear gradient from 5% to 25% of MeCN (25 min), 95% MeCN (30 min)] and UV detection at 220 nm.

## Nuclear magnetic resonance

[1]H spectra were acquired at 300K with an Agilent UNITY INOVA spectrometer operating at a Larmor frequency of 500 MHz. Spectra were acquired with a pulse of 6.1 μs (90°), 5 s recycle delay, 48 transients over a spectral window of 6000 Hz. Homonuclear [1]H gCOSY was acquired with the same acquisition parameters and sampling each of the 512 increments with 64 transients and 2048 complex points. The same parameters were applied for homonuclear [1]H ROESY experiments with both 200 and 300 ms mixing time and applying the MLEV17 scheme for the spin-lock. The [1]H chemical shift scale was referenced to the solvent residual resonance.

Titrations were performed in acetate buffered saline (ABS) at pH five prepared in $D_2O$ (pD 5.4), at constant ligand concentration in the 2–3 mM range by varying metal concentration ($Ga^{3+}$), which

**Table 2.** Crystallographic data and refinement statistics

| | Semet 6H7V | Apo 6HCP | Complex 6H7F |
|---|---|---|---|
| Data collection | | | |
| Space group | C 1 2 1 | C 1 2 1 | C 1 2 1 |
| Cell dimensions | | | |
| $a$, $b$, $c$ (Å) $\alpha$, $\beta$, $\gamma$ (°) | 182.3, 220.4, 101.7 90, 98.7, 90 | 180.2, 219.5, 101.4 90, 99.2, 90 | 179.6, 220.7, 101.1 90, 99.2, 90 |
| Resolution (Å) | 139.53–2.54 (2.61–2.54)* | 57.24–1.83 (1.88–1.83)* | 49.93–2.26 (2.30–2.26) |
| $R_{sym}$ or $R_{merge}$ | 0.12 (0.789) | 0.049 (0.665) | 0.108 (0.844) |
| $I / \sigma I$ | 19.3 (3.8) | 17.1 (2.2) | 9.2 (1.6) |
| Completeness (%) | (99.9) 100 | 99.7 (99.6) | 99.8 (99.9) |
| Redundancy | 15.2 (14.1) | 3.8 (3.9) | 3.8 (3.8) |
| CC half | - | - | 0.995 (0.527) |
| Anom completeness | 99.9 (99.9) | - | - |
| Ano multiplicity | 7.5 (6.9) | - | - |
| Refinement | | | |
| Resolution (Å) | 139.53–2.54 | 57.24–1.83 | 49.93–2.26 |
| No. of reflections | 123,580 | 322,409 | 171,799 |
| $R_{work}$/$R_{free}$ | 0.183/0.213 | 0.156/0.176 | 0.187/0.217 |
| No. of atoms | | | |
| Protein | 15,424 | 16,060 | 15,854 |
| Ligand/ion | 201 | 754 | 691 |
| Water | 303 | 1995 | 752 |
| B-factors | | | |
| Protein | 44.4 | 33.40 | 39.70 |
| Ligand/ion | 60.4 | 57.70 | 59.22 |
| Water | 36.90 | 45.90 | 36.87 |
| R.m.s deviations | | | |
| Bond lengths (Å) | 0.010 | 0.008 | 0.008 |
| Bond angles (°) | 1.24 | 1.16 | 0.958 |

Each dataset was collected from a single crystal. *Values in parentheses are for highest-resolution shell.

DOI: https://doi.org/10.7554/eLife.42270.017

was added in the form of $Ga(ClO_4)_3$. Relative intensities of selected resonances from the same proton group in the different species formed along the titration were measured and normalized for each spectrum. Normalized NMR intensities were analysed by fitting model equations through a Monte-Carlo minimization scheme. The following stepwise equilibria were taken into consideration at constant pH:

$$L + L_{n-1}M \rightleftharpoons L_nM \quad ; \quad K_n = \frac{[L_nM]}{[L][L_{n-1}M]} \tag{1}$$

where L is the ligand, M the metal, $K_n$ is the apparent equilibrium constant, n is an integer from 1 to 3, and molar concentrations are indicated with brackets. In addition to these mass-action laws, the following mass-balance relationships must also be applied:

$$[L_0] = [L] + \sum_{n=1}^{3} n \cdot [L_nM] \tag{2}$$

$$[M_0] = [M] + \sum_{n=1}^{3}[L_nM] \tag{3}$$

where $[L_0]$ and $[M_0]$ are the total ligand and total metal molar concentration, respectively. Finally, the following quartic equation is obtained for $[L]$, that is the free ligand at equilibrium:

$$K_1K_2K_3[L]^4 + (K_1K_2 - K_1K_2K_3[L_0] + 3K_1K_2K_3[M_0])[L]^3 \\ + (K_1 - K_1K_2[L_0] + 2K_1K_2[M_0])[L]^2 + (1 - K_1[L_0] + K_1[M_0])[L] - [L]_0 = 0 \tag{4}$$

which was solved numerically with the Newton-Raphson method. A Monte-Carlo scheme was implemented to minimize the sum of square difference between the experimental and calculated data points, by using the $K_n$ as fitting parameters. One parameter was randomly selected at each iteration and randomly varied between $-10\%$ and $+10\%$. Only the steps resulting in a decrease of the sum of square difference were accepted, rejecting the last move otherwise. All the datasets pertaining to the different species along the same titration were fitted simultaneously.

## Acknowledgements

We thank Basilea Pharmaceutica Ltd as partner of the IMI, Eric Desarbre for the synthesis of acinetobactin and preacinetobactin, Sophie Guillard for the analytical analysis she did to confirm the structures of the substances and A Tortajada for the technical help.

The research leading to these results was conducted as part of the Translocation consortium (www.translocation.eu) and benfitted from support from ND4BB ENABLE Consortium which has received support from the Innovative Medicines Initiatives Joint Undertaking under Grant Agreement Nr 115525 and Nr 115583, resources which are composed of financial contribution from the European Union's seventh framework programme (FP7/2007–2013) and EFPIA companies in kind contribution. This is work is supported by a Wellcome Trust Investigator (100209/Z/12/Z) award.

## Additional information

### Funding

| Funder | Grant reference number | Author |
| --- | --- | --- |
| European Commission | 115525 | Lucile Moynie<br>Ilaria Serra<br>Mariano A Scorciapino<br>Malcolm GP Page<br>Matteo Ceccarelli<br>James H Naismith |
| Wellcome Trust | 100209/Z/12/Z | Lucile Moynie<br>James H Naismith |
| European Commission | 115583 | Lucile Moynié<br>James H Naismith |

The funders had no role in study design, data collection and interpretation, or the decision to submit the work for publication.

### Author contributions

Lucile Moynié, Conceptualization, Formal analysis, Supervision, Funding acquisition, Investigation, Methodology, Writing—original draft, Project administration, Writing—review and editing; Ilaria Serra, Formal analysis, Investigation, Methodology, Writing—original draft, Writing—review and editing; Mariano A Scorciapino, Emilia Oueis, Formal analysis, Methodology; Malcolm GP Page, Methodology, Writing—review and editing; Matteo Ceccarelli, Conceptualization, Methodology, Writing—original draft, Project administration, Writing—review and editing; James H Naismith, Conceptualization, Formal analysis, Supervision, Funding acquisition, Investigation, Writing—original draft, Project administration, Writing—review and editing

Author ORCIDs
Lucile Moynié http://orcid.org/0000-0002-4097-4331
Emilia Oueis http://orcid.org/0000-0002-0228-6394
James H Naismith http://orcid.org/0000-0001-6744-5061

## Decision letter and Author response

Decision letter https://doi.org/10.7554/eLife.42270.026
Author response https://doi.org/10.7554/eLife.42270.027

## Additional files

### Data availability

Diffraction data have been deposited in PDB under the accession codes 6H7F, 6H7V, 6HCP.

The following datasets were generated:

| Author(s) | Year | Dataset title | Dataset URL | Database and Identifier |
|---|---|---|---|---|
| Moynie L, Naismith JH | 2018 | Crystal structure of BauA, the Ferric preacinetobactin receptor from Acinetobacter baumannii in complex with Fe3+-Preacinetobactin-acinetobactin | https://www.rcsb.org/structure/6H7F | Protein Data Bank, 6H7F |
| Moynie L, Naismith JH | 2018 | Crystal structure of BauA, the Ferric preacinetobactin receptor from Acinetobacter baumannii | https://www.rcsb.org/structure/6H7V | Protein Data Bank, 6H7V |
| Moynie L, Naismith JH | 2018 | Crystal structure of BauA, the Ferric preacinetobactin receptor from Acinetobacter baumannii | https://www.rcsb.org/structure/6HCP | Protein Data Bank, 6HCP |

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
