## [Decision Letter]

Thank you for submitting your article "Preacinetobactin not acinetobactin is essential for iron uptake by the pathogen *Acinetobacter baumannii*" for consideration by *eLife*. Your article has been reviewed by three peer reviewers, including Wilfred A van der Donk as the Reviewing Editor and Reviewer #1, and the evaluation has been overseen by Gisela Storz as the Senior Editor. The following individual involved in review of your submission has agreed to reveal their identity: Timothy Wencewicz (Reviewer #2).

The reviewers have discussed the reviews with one another and the Reviewing Editor has drafted this decision to help you prepare a revised submission.

Summary:

The authors describe a novel structure of BauA, the TonB-dependent siderophore outer membrane receptor from *Acinetobacter baumannii*, in complex with a novel mixed ligand siderophore complex composed of one iron, one pre-acinetobactin, and one acinetobactin. The new structure is important from several perspectives 1) it gives the first structural glimpse of a TonB-dependent siderophore receptor from *A. baumannii*, 2) it gives the first structural evidence that pre-acinetobactin is recognized by BauA which provides clarity on a literature debate for the biologically relevant isomer, pre-acinetobactin or acinetobactin, and 3) the chelation mode of pre-acinetobactin is revealed showing occupancy of four coordinate sites by pre-acinetobacter and leaving two coordination sites for the catecholate of acinetobactin. The data also provides information regarding potential cargo attachment for siderophore-antibiotic conjugates. The crystallographic data are supported by binding studies.

Essential revisions:

1) There is some recent consensus in the literature that there are two ligand binding sites in these types of receptors, one peripheral (secondary) and one deeper inside the protein (primary). Is this the case for BauA, and if so, which binding site is occupied here? From the structure it appears that the ligands make very few interactions with the plug domain, and this raises the question if and how occupation of this binding site causes a conformational change of the N-terminus of the plug, including the TonB box. Are there any differences with the plug of the apo structure? This entire issue is important, because if the observed binding site is not the primary one, the main selling point, i.e. relevance of the structure for antibiotic delivery is not clear.

2) It is not demonstrated which of the two siderophores is actually transported. Literature evidence suggests that both siderophores can serve as iron sources for wild type and pre-acinetobactin biosynthetic incompetent mutants of *A. baumannii* (Shapiro and Wencewicz, 2016). Indeed, acinetobactin could become relevant or even essential for import downstream of BauA. It is also possible that there is a second TonB-dependent transporter for the acinetobactin ferric complex. Such alternative scenarios should be discussed and the title should be revised because the biological relevance for acinetobactin is not fully ruled out. A possible title could be something like "Preacinetobactin is essential for iron uptake by the pathogen *Acinetobacter baumannii*".

3) The authors should show the NMR data currently presented in Table 1. They should also show the NMR data for the 1:1 and 1:2 complexes with metal since this is essential information. Furthermore, which resonances were used for Figure 3B? It seems that the equation used for the fit expects that at higher and higher metal equivalents the material is driven to the LM complex, but the data seem to indicate that a final equilibrium is reached between LM and L_2_M. Did the authors consider other models? It does not seem that the LM complex goes away at high metal equiv. The authors should have a few more data points at 1.4 and 1.5 equivalents to see if this is indeed the case, and if not should either explain this or consider another model.

4) It is surprising that acinetobactin is obtained as a mixture of diastereomers. No details on the synthesis are given and epimerization at this carbon was not observed in Shapiro and Wencewicz, 2016; Song et al., 2017; Shapiro and Wencewicz, 2017 and Takeuchi et al., 2010. What stereoisomer of Thr was used in the synthesis? Why/how did the epimerization happen? Normally, methyl oxazolines are resistant to epimerization. Since other studies do not report this epimerization, more information should be provided. Furthermore, is it possible that the mixed electron density is due to a mixture of pre-acinetobactin and acinetobactin at this position in the crystal structure, and not diastereomers? In the Discussion the authors first mention that an incorrect diastereomer is formed in the chemical synthesis (it should be mentioned much earlier when the diastereomers are observed), but no reference is provided here.

---

## [Author Response]

Essential revisions:1) There is some recent consensus in the literature that there are two ligand binding sites in these types of receptors, one peripheral (secondary) and one deeper inside the protein (primary). Is this the case for BauA, and if so, which binding site is occupied here?

We do not believe there is any evidence for a second site in BauA (but are aware of the debate), our ITC fits best with a single recognition site (Figure 3—figure supplement 2). We show in Figure 1—figure supplement 1 that BauA has a much more open structure. If one compares the position of Fe relative to secondary structure (plug and barrel) of other receptor complexes, we see Fe is in a similar position (Figure 2—figure supplement 1). We say “in neither case did we see any evidence for a biphasic recognition model that has been proposed between enterobactin and FepA (Payne et al., 1997).”

From the structure it appears that the ligands make very few interactions with the plug domain, and this raises the question if and how occupation of this binding site causes a conformational change of the N-terminus of the plug, including the TonB box.

There are indeed fewer hydrogen bonds, but the ligand is hydrophobic, we now note: “Although there are no hydrogen bonds between the siderophores and the plug domain, two residues of the plug domain (Trp97 and Tyr125) do make van der Waal interactions with preacinetobactin.”

Are there any differences with the plug of the apo structure?

We do not see any and this is noted. “There are no changes in the TonB region as a result of ligand binding.”

This entire issue is important, because if the observed binding site is not the primary one, the main selling point, i.e. relevance of the structure for antibiotic delivery is not clear.

We agree this is important and we believe the ITC and specific interactions revealed show this is the correct binding site. Further we note in Figure 2—figure supplement 1, the siderophore is located in the “right” position at the plug domain.

2) It is not demonstrated which of the two siderophores is actually transported. Literature evidence suggests that both siderophores can serve as iron sources for wild type and pre-acinetobactin biosynthetic incompetent mutants of A. baumannii (Shapiro and Wencewicz, 2016). Indeed, acinetobactin could become relevant or even essential for import downstream of BauA. It is also possible that there is a second TonB-dependent transporter for the acinetobactin ferric complex. Such alternative scenarios should be discussed and the title should be revised because the biological relevance for acinetobactin is not fully ruled out. A possible title could be something like "Preacinetobactin is essential for iron uptake by the pathogen Acinetobacter baumannii".

We agree and have changed the title to “Preacinetobactin not acinetobactin is essential for iron uptake by the BauA transporter of the pathogen *Acinetobacter baumannii*”.

We note in the Discussion “Our work does not eliminate the possibility of another TBDT transporting acinetobactin, but so far none has been identified in *A. baumannii*.”

3) The authors should show the NMR data currently presented in Table 1. They should also show the NMR data for the 1:1 and 1:2 complexes with metal since this is essential information.

A new figure (Figure 3—figure supplement 3 in the revised version) has been added where both preacinetobactin and acinetobactin spectra are shown together with labels for the different resonances. Assignments for the free ligand, the ML_2_ and the ML complexes are shown.

Furthermore, which resonances were used for Figure 3B?

It was the resonance due to the methyl group. This information has been included in the revised caption.

It seems that the equation used for the fit expects that at higher and higher metal equivalents the material is driven to the LM complex, but the data seem to indicate that a final equilibrium is reached between LM and L_2_M. Did the authors consider other models? It does not seem that the LM complex goes away at high metal equiv. The authors should have a few more data points at 1.4 and 1.5 equivalents to see if this is indeed the case, and if not should either explain this or consider another model.

The reviewer makes an interesting point at high metal concentrations. We agree that in the simple model the L_2_M is expected to approach zero concentration with increasing the metal concentration. The presence of the 10 to 20% acinetobactin in the sample (relative to preacinetobactin concentration, acinetobactin is visible in the spectrum at 0 eq. Ga^3+^, Figure 3—figure supplement 3) could play a role where metal is in excess. Acinetobactin would coordinate the metal ion some of Ga preacinetobactin (ML complex), to form preacinetobactin-acinetobactin mixed complexes (MLL^*^) seen in the crystal structure. (Acinetobactin would not form the ML type complexes when preacinetobactin is present). The NMR of the preacinetobactin in the mixed complexes would be practically indistinguishable from the purely preacinetobactin ML_2_ complexes. Consequently, the integrated resonances for ML_2_ would have the contribution from MLL^*^ complexes thus masking at high metal concentrations the disappearance of ML_2_. Curve-fitting to the available data points was very poor and cannot be used to support this hypothesis. Finally, we observed that spectra quality decreased along the titration so we focused our discussion on the regime below 1 eq. Ga^3+^, which we felt was more physiologically relevant.

It is clear that acinetobactin does not play any role in this lower concentration regime and obtained a good fit in this regime with the model.

We say: “Beyond 1.0 eq. Ga^3+^, a noticeable deviation between the model and the experimental data can be seen (Figure 3B). […] Preacinetobactin in these mixed complexes would be magnetically undistinguishable from those in the ML_2_ complexes, thus artificially inflating the concentration of this species.”

4) It is surprising that acinetobactin is obtained as a mixture of diastereomers. No details on the synthesis are given and epimerization at this carbon was not observed in Shapiro and Wencewicz, 2016; Song et al., 2017; Shapiro and Wencewicz, 2017 and Takeuchi et al., 2010. What stereoisomer of Thr was used in the synthesis? Why/how did the epimerization happen? Normally, methyl oxazolines are resistant to epimerization. Since other studies do not report this epimerization, more information should be provided. Furthermore, is it possible that the mixed electron density is due to a mixture of pre-acinetobactin and acinetobactin at this position in the crystal structure, and not diastereomers? In the Discussion the authors first mention that an incorrect diastereomer is formed in the chemical synthesis (it should be mentioned much earlier when the diastereomers are observed), but no reference is provided here.

We agree that the second catecholate binding molecule was surprising. It is not central to the paper’s message, since protein recognises preacinetobactin but it is a puzzle. We have no good explanation for the density we observe for the second catecholate molecule except the presence other diastereomer. The molecule was a gift and made following literature now cited. The NMR of the material (Figure 3—figure supplement 5) matches the literature and shows no additional significant peaks, so we conclude that the sample is essentially pure. We agree with the reviewer, chemical interconversion seems improbable.

Yet, we come to our data Figure 2—figure supplement 2, we show the refinement of the second molecule as:

A) preacinetobactin correct diastereomer; clearly does not fit we get a large negative peak for the preacinetobactin molecule. Given the conditions, such catechol only ligated preacinetobactin would be predicted to have converted to acinetobactin.

B) preacinetobactin acinetobactin mixture, both correct diastereomer, we get a large negative peak for the preacinetobactin molecule

C) acinetobactin correct diastereomer, the density largely fits, however there is residual additional density

D) acinetobactin wrong diastereomer, the density fits, however there is significant residual additional density

E) acinetobactin both diastereomers, the density fits, no residual additional density

We include some new data that was not included in the original manuscript (panels F and G). We had obtained a lower resolution co crystal structure with “pure” acinetobactin. The density for this was even clearer (in our view) as both diastereomers.

We say: “The experimental electron density is most convincingly fitted by a mixture of two diastereomers of acinetobactin (Figure 2C), in an earlier co-crystallisation experiment the “wrong” diastereomer of acinetobactin was clearer. […] Thus, we are left to suggest the presence of small contaminating amount of the “wrong” diastereomer or chemical conversion under crystallisation conditions (protein, buffer, salts, crystal effects etc.).”

Although unlikely, we accept that our interpretation of the density for this second molecule may be wrong (although as we show other alternatives look even worse). We decided to show the data (rather than hide the presence of what we think is the “wrong” diastereomer) and give our best interpretation. Since it is side issue to main message, we felt this was the most acceptable way forward.